# Cherry Polyphenol Extract Ameliorated Dextran Sodium Sulfate-Induced Ulcerative Colitis in Mice by Suppressing Wnt/β-Catenin Signaling Pathway

**DOI:** 10.3390/foods11010049

**Published:** 2021-12-26

**Authors:** Fuhua Li, Huiming Yan, Ling Jiang, Jichun Zhao, Xiaojuan Lei, Jian Ming

**Affiliations:** 1College of Food Science, Southwest University, Chongqing 400715, China; fuhualee92@163.com (F.L.); yhm2639722584@163.com (H.Y.); 15823556360@163.com (L.J.); jichunzhao@swu.edu.cn (J.Z.); xjuanlei@swu.edu.cn (X.L.); 2Research Center of Food Storage & Logistics, Southwest University, Chongqing 400715, China

**Keywords:** ulcerative colitis, cherry polyphenol, anti-inflammatory, antioxidant, Wnt/β-catenin signaling pathway

## Abstract

Ulcerative colitis (UC) is a chronic and nonspecific inflammatory disease of the colon and rectum, and its etiology remains obscure. Cherry polyphenols showed potential health-promoting effects. However, both the protective effect and mechanism of cherry polyphenols on UC are still unclear. This study aimed to investigate the potential role of the free polyphenol extract of cherry in alleviating UC and its possible mechanism of action. Our study revealed that the free polyphenol extract of cherry management significantly alleviated UC symptoms, such as weight loss, colon shortening, the thickening of colonic mucous layer, etc. The free polyphenol extract of cherry treatment also introduced a significant reduction in levels of malondialdehyde (MDA), myeloperoxidase (MPO) and nitric oxide (NO), while causing a significant elevation in levels of catalase (CAT), glutathione (GSH-Px), superoxide dismutase (SOD), as well as the downregulation of pro-inflammatory cytokines. This indicated that such positive effects were performed through reducing oxidative damage or in a cytokine-specific manner. The immunofluorescence analysis of ZO-1 and occludin proteins declared that the free polyphenol extract of cherry had the potential to prompt intestinal barrier function. The reduced expression levels of β-catenin, c-myc, cyclin D1 and GSK-3β suggested that the cherry extract performed its positive effect on UC by suppressing the Wnt/β-ctenin pathway. This finding may pave the way into further understanding the mechanism of cherry polyphenols ameliorating ulcerative colitis.

## 1. Introduction

Ulcerative colitis (UC) is chronic and nonspecific inflammatory disease of the colon and rectum, with typical symptoms such as blood in the stool, diarrhea, incontinence, mucus discharge, and nocturnal defecations, etc. [1,2,3]. Some medications (corticosteroids, aminosalicylic acid and adrenal glucocorticoids) were employed to treat UC, but most had side effects; these have always been the main clinical problems [4,5,6,7]. Alternatively, dietary polyphenols were reported to have the potential to alleviate the severity of UC symptoms [8,9].

The etiology of UC remains obscure, although it is generally considered that immunological, genetic, and environmental factors, including diet and free radicals could trigger UC [10,11]. It is widely accepted that the occurrence and development of UC were linked to many factors, that is, the production of reactive oxygen species (ROS) and pro-inflammatory cytokine [12], the loss of tight junction (TJ) proteins [13], the imbalance of Th1 and Th2 cells [14], and changes in intestinal permeability and regeneration ability [15], as well as the imbalance between transglutaminase (TG) and calpain gene [2]. Among all the listed factors, ROS or other pro-inflammatory cytokines were the initiators, since they were involved in triggering and aggravating the disruption of intestinal barrier integrity [15]. Polyphenols with great antioxidant and anti-inflammatory properties were, therefore, appreciated for their potential to reduce the severity of UC. Polyphenols contributed to reducing the damage of ROS to gastrointestinal mucosa [16], also enhancing the intracellular “self-defense” to fight against oxidative stress and inflammation [17]. However, the further mechanism of action linked to such a positive effect of polyphenols on intestinal tract is still unclear.

Cherry possess high amounts of polyphenols and natural antioxidants, e.g., flavononoids, phenolic acids, and anthocyanins [18,19]. In particularly, cherry polyphenols showed anti-inflammatory properties [20], a high antioxidant activity [21], and other potential health-promoting effects [22]. However, the protective effect and mechanism of cherry polyphenols on UC are still unclear.

Here, we explored the effect of the polyphenol extract from cherry on the colon (length and histological damage), activity of antioxidant enzymes, levels of antioxidant and inflammatory factors, as well as tight junction proteins in UC mice induced by dextran sodium sulfate (DSS). The possible mechanism of action was also studied from the point of the Wnt/β-ctenin pathway. For a further understanding of the composition of polyphenols extract administrated to UC mice, we identified the cherry polyphenols by UPLC-ESI-TOF-MS/MS.

## 2. Materials and Methods

### 2.1. Materials

We used “Meizao” cherry (*Cerasus pseudocerasus* (Lindl.) G. Don, (Prunus pseudocerasus Lindl.)). We purchased Meizao from Yantai cherry base (Shandong province, China). LC-MS grade acetonitrile was purchased from Sigma (St. Louis, MO, USA). Dextran sodium sulfate (DSS, 36,000–50,000 MW) was purchased from Aladdin Biomedicals (Chongqing, China). Commercially available kits for alanine aminotransferase (ALT), catalase (CAT), glutathione (GSH-Px), superoxide dismutase (SOD), malondialdehyde (MDA), myeloperoxidase (MPO) as well as nitric oxide (NO) and commercial enzyme-linked immunosorbent assay (ELISA) kits for determining IL-6, TNF-α and IL-10 were obtained from Nanjing Jancheng Bioengineering Institute (Nanjing, China). Salazosulfapyridine (SASP) was purchased from Xinyi Tianping Pharmaceutical Co., Ltd. (Shanghai, China).

Occludin and ZO-1 antibodies were purchased from Bioss (Beijing, China) and Affinty (Jiangsu province, China), respectively. DAB concentration Kits were obtained from Zhongshan Jinqiao biology Co., Ltd. (Beijing, China). Protein extraction kits, BCA protein assay kits, antibodies against β-catenin, c-myc, Cyclin D1, GSK-3β and β-actin were purchased from Servicebio technology CO., LTD (Wuhan, China).

### 2.2. Extraction, Purification and Content Determination of the Cherry Polyphenol Extracts

The extraction was performed according to the studies [23,24]. Briefly, fresh cherries were artificially enucleated to obtain the edible part, which was then freeze-dried in vacuum and broken into powder manually. The powder was mixed with 80% cold acetone. The mixture was then homogenized and centrifuged (4 ℃, 10 min, 3500 r/min). The supernatant was collected and evaporated at 45 ℃ to obtain the crude extract of free polyphenol. The residue from the free extract was re-extracted to obtain the bound polyphenol extract of cherry, and this procedure was performed according to the literature [23]. Briefly, 20 mL 2 mol/L NaOH solution was added to the residue. After reaction for 1.5 h, the pH of the mixture was adjusted to 2.0 with hydrochloric acid (12 M). Hexane (20 mL) was added for removing lipids. The bound polyphenol was extracted five times by 100 mL of ethyl acetate. Then, the ethyl acetate part was collected and evaporated, and reconstituted in chromatographic methanol to obtain the bound polyphenol extract of cherry. The free and bound extracts were stored at −80 °C.

The purification of the crude extract of free polyphenol was performed according to the previous reports [25,26]. The AB-8 macroporous resin was used to remove impurities, i.e., polysaccharides, proteins, etc. The AB-8 macroporous resin was pretreated according to the instructions, and then it was loaded into the chromatography column (ф 2.5 × 40 cm). The elution parameters were as follows: 4 BV column volume of distilled water was used to balance the column; the loading velocity was 15 rpm; the elution flow rate was 10 rpm; and the eluent was 60% ethanol aqueous solution (pH = 3). All eluents were collected, concentrated under pressure, and lyophilized in vacuum to obtain the purified free polyphenol extract.

The contents of total polyphenols in samples were analyzed by the Folin–Ciocalteu colorimetric method described previously [23]. Gallic acid was the standard, and the regression equation for the standard curve of gallic acid was y = 0.0039x + 0.0318 (*R*^2^ = 0.9976), and the result was expressed as mg of gallic acid equivalent per gram of edible part of dry cherry (mg GAE/g DW).

### 2.3. The Antioxidant Capacities In Vitro of the Free and Bound Polyphenol Extracts

The antioxidant capacities of the cherry free and bound polyphenols extract were estimated by several assays including 2,2-diphenyl-1-picrylhydrazyl (DPPH), 2,2-azinobis (3-ethyl-benzothiazoline-6-sulfonic acid) (ABTS), ferric-reducing antioxidant power (FRAP), and the oxygen radical absorption capacity (ORAC). The ABTS, DPPH, FRAP, and ORAC assays were performed according to the literature [27]. The results of the ABTS, DPPH, and FRAP assays were expressed as the EC_50_ value (μg/mL), and the ORAC value was expressed as μmol Trolox equivalent per gram of edible part of dry cherry (μmol TE/g DW).

### 2.4. Analysis of Cherry Polyphenol Extracts (Free and Bound Polyphenols) by HPLC and UPLC-ESI-TOF-MS/MS

The identification of free and bound polyphenols was performed according to the report [28,29,30]. For the quantitative chromatographic analysis by HPLC, a thermo BDS C_18_ chromatographic column (250 × 4.6 mm, 5 μm) was used for separation: mobile phase A was 0.1% formic acid aqueous solution, and the mobile phase B was 100% acetonitrile. The column temperature was set at 40 °C with a flow rate 0.7 mL/min and injection volume of 10 μL. The gradient elution procedure was 0–5 min, 90%A; 5–50 min, 90–60%A; 50–55 min, 60–10%A; 55–62 min, 10%A; 62–65 min, 10–90%A; and 65–75 min, 90%A.

The UPLC-ESI-TOF-MS/MS analysis was carried out using an LC-30AD ultra performance liquid chromatography (Shimadzu, Kyoto, Japan) equipped with a SPD-M20A photodiode array detector (Shimadzu, Kyoto, Japan), with a mass detector Triple TOF^TM^4600 micro mass spectrometer (AB Sciex^TM^, Milwaukee, Wisconsin, USA) equipped with an electrospray ionization (ESI) source operating in negative and positive modes. Separations of individual polyphenols were carried out using a ZORBAX Eclipse XDB-C_18_ column (150 × 2.1 mm, 1.8 μm, Agilent, Santa Clara, CA, USA) at 30 °C. The mobile phase consisted of solvent A (ultrapure water) and solvent B (100% acetonitrile). The gradient elution procedure was 0–1 min, 10%B; 1–2.5 min, 10~25%B; 2.5–8.5 min, 25~95%B; 8.5–12 min, 95%B; and 12–12.1 min, 95–10%B. The isocratic flow rate was 0.3 mL/min with 3 μL of injection volume. MS^n^ detection parameters were: positive ESI mode; 5.5 kV capillary voltage; 100–1000 *m/z* scan range; both the nebulizer (air) and auxiliary (air) flow rates were set at 55 psi, the curtain gas (nitrogen) was set at 25 psi. The temperature of the heater was 600 °C, the resolution was 30,000, and the collision energy (CE) was 10 V.

### 2.5. Experimental Animals

BALB/c mice (*n* = 50, male, 6–8 weeks old, body weight 17 ± 2 g) were purchased from Byrness Weil biotech Ltd. (Chongqing, China). Experimental mice were allowed to acclimate for one week in a temperature-controlled room (25 ± 1 °C), under a 12 h light/dark cycle in SPF animal room of Animal Center, School of Pharmacy, Southwest University, Chongqing, with free access to drinking water and standard chow. The experimental procedures were approved by the Animal Care and Use Committee of Southwest University in China, and the IACUC Issue No. is IACUC-20200315-02.

Mice were randomly separated into five groups by weight (*n* = 10/group). Before modeling, the protection of colon was performed in advance for two days (once daily, at a similar time of the day), that is, the free polyphenol extract of cherry was dissolved in pure water. Gavage administration was performed to feed mice. Mice in the high and low dose group were fed 450 mg/kg BW and 150 mg/kg BW of the free polyphenol extract of cherry, respectively. Mice in the positive group were fed 300 mg/kg BW SASP, and the normal and control mice were administrated with equal volume of drinking water. Here, the doses of cherry polyphenol extract were calculated based on the daily intake of polyphenols in human diets by using body surface area method [31,32,33,34]. The choice of the SASP dose (300 mg/kg BW) was first based on the daily intake amount of SASP (2–3 g), Then, such a daily intake for humans was converted to the dose for mice according to the ratio (0.0026) of human body surface area to mouse body surface area, as well as the body weight of the tested mice.

### 2.6. Developing of Ulcerative Colitis and Cherry Polyphenol Extract Treatment

Ulcerative colitis was induced with DSS according to the method reported by Zhang et al. [35]. The following groups were included: the normal group (N), the control group (C), the positive control group (P), the high dose group (H) and the low dose group (L). N mice were treated with drinking water. C mice were fed with DSS solution (2.5%, *w/v*) via their drinking water for 14 consecutive days for ulcerative colitis induction. P mice were fed with 300 mg/kg BW SASP by gavage once a day and received 2.5% DSS in their drinking water for consecutive 14 days. H mice were fed with the free polyphenol extract of cherry at dose of 450 mg/kg BW by gavage once a day and received 2.5% DSS in their drinking water for 14 consecutive days. L mice were fed with the free polyphenol extract of cherry at dose of 150 mg/kg BW by gavage once a day and received 2.5% DSS in their drinking water for 14 consecutive days. The body weight, food intake and fecal status were routinely monitored during the experiment. The disease activity index (DAI) of each group was calculated based on the body mass, stool shape and bloody situation of stool according to the previous report [36]. The mice were fasted for 12 h before they were sacrificed, then the blood, colon tissues and other viscera were collected and stored at −80 °C.

### 2.7. Histological Assessment

A histological assessment was carried out according to the reported method [37]. Briefly, colon was firstly washed with PBS, then the one-centimeter section from the distal end of colon was cut, fixed with 4% paraformaldehyde, and then embedded in paraffin. The paraffin-embedded colon tissue was sectioned and stained with hematoxylin and eosin (H&E). The morphological changes of intestinal epithelium and submucosal infiltration were each scored on a 0–4 scale of severity. The final total histological score (from 0 to 8) was determined by the sum of the two parameters above.

### 2.8. Immunohistochemistry Analysis

An immunohistochemistry analysis was performed according to the reported method [35]. Briefly, the paraffin-embedded distal colon (5 μm thick) was incubated with occludin and ZO-1 antibody overnight, and then incubated with a biotinylated secondary antibody for 30 min. Signals were visualized using DAB concentration kits and hematoxylin counterstaining. BA200Digital microscope (400×) was used for images.

### 2.9. Determination of Enzymes in Serum

The activity of enzymes including ALT, CAT, GSH-Px, SOD and MDA in serum was measured by using the commercially available kits, and the testing process was strictly in accordance with the provisions of the instructions.

### 2.10. Determination of Inflammatory Cytokines in the Colon

The determination of inflammatory cytokines in the colon was carried out according to the reported method [38]. Briefly, colon tissue was homogenized and centrifuged at 4 °C to collect the supernatant. The levels of IL-6, TNF-α, IL-10 in the supernatant were quantified by a commercial, enzyme-linked immunosorbent assay (ELISA) kit. MPO and NO contents were determined by commercially available kits following the manufacturer’s instructions. The concentration of protein was determined with the Bradford assay according to the manual instructions.

### 2.11. Western Blot Analysis

A Western blot analysis was carried out according to the reported method [39,40]. Total protein extracts were subjected to SDS-PAGE and electro transferred to a nitrocellulose membrane. The membrane was decolorized and sealed with 5% skim milk (in 0.5% TBST) for one hour. The primary antibodies were diluted (5% skim milk dissolved in TBST) and incubated at 4 °C. After being washed by TBST, membranes were incubated with horseradish peroxidase (HRP)-conjugated anti-goat, anti-mouse, or anti-rabbit secondary antibody, and then developed using the ECL detection system (Servicebio) according to the manufacturer’s instructions.

### 2.12. Statistical Analysis

All data were presented as mean ± standard deviation (SD). Differences were analyzed by one-way analysis of variance (ANOVA) and Duncan’s Multiple Comparison Test using SPSS 20.0 software (IBM, Armonk, NY, USA). A *p*-value of less than 0.05 was considered significant.

## 3. Results

### 3.1. The Polyphenols Content and Antioxidant Capacities In Vitro of the Free and Bound Polyphenol Extract

Phenolic compounds are commonly combined with food matrix (proteins, dietary fiber, etc.) by hydrogen bonding, hydrophobic interaction or covalent bonding, and they can be divided into the soluble-free phenolics and bound phenolics according to how close they were combined. The former was conducted on the basis of solvent-soluble extraction, and it contained both free aglycones and soluble conjugates (glycosylated forms); the bound phenolics were and mainly in the form of β-glycosides that were difficult to extract [23]. We analyzed both the free and bound polyphenol extracts of cherry. As shown in Appendix A, the total polyphenol content (free + bound) of the cherry extract, tested by the Folin–Ciocalteu method, was about 6.84 mg GAE/g DW, and the polyphenol content of the free extract was approximately 50-fold higher than that of the bound extract. The free polyphenol extract also showed lower EC_50_ values indicating higher antioxidant capacities, as evaluated by DPPH and FRAP assays. The ORAC value of the free polyphenol extract was 36-fold higher than that of the bound polyphenol extract. However, the bound polyphenol extract presented a higher scavenging ability against ABTS free radicals due to its lower EC_50_ value than that of the free polyphenol extract. The difference in the antioxidant capacities of free and bound polyphenol extracts indicated that the polyphenols composition of the free and bound extract might be different. Therefore, we analyzed the composition of the free and bound polyphenol extract.

### 3.2. The Quantitative Analysis of Phenolic Compounds in the Cherry Extract by HPLC

The phenolic composition of the free and bound extract was first analyzed by HPLC, and phenolic compounds were identified by a comparison with both the maximum UV absorption wavelength and the retention time of the phenolic standards. As shown in Appendix A, the most abundant component of the free polyphenol extract was cyanidin 3-*O*-glucoside (1276.48 μg/g DW) followed by rutin (177.44 μg/g DW) and chlorogenic acid (153.81 μg/g DW). However, those polyphenols were not detected in the bound extract. Generally, the number of the identified phenolic compounds by HPLC was very small, and many more details should be provided to support the phenolic composition of the cherry extracts. Therefore, UPLC-ESI-TOF-MS/MS was performed for further analysis.

### 3.3. Identification of Cherry Polyphenol Extract by UPLC-ESI-TOF-MS/MS

Both of the free and bound polyphenol extracts were analyzed by UPLC-ESI-TOF-MS/MS for further understanding the phenolic composition of the cherry extracts. All these compounds were preliminarily identified based on the information on the parent ions and their fragment ions, as well as the published reports. The secondary mass spectra of the identified polyphenols are shown in Appendix A. The mass spectrum data and corresponding studies to support the identified polyphenols in the extract are summarized in Table 1.

A total of 15 compounds were preliminarily identified in the free polyphenol extract, including five anthocyanins, one hydroxybenzoic acid, four hydroxycinnamic acids, four flavonols and one flavanol. The number of the identified compounds in the bound polyphenol extract was less than that of the free extract, and there were only two anthocyanins, one hydroxycinnamic acid and one flavonol in the bound polyphenol extract.

Above all, the free polyphenol extract was likely higher in both its number of polyphenol compounds and contents. Therefore, only the free polyphenol extract of cherry was evaluated for its effect on ulcerative colitis in mice.

### 3.4. Effects of the Free Polyphenol Extract of Cherry on DSS-Induced Ulcerative Colitis in Mice

#### 3.4.1. Effect of the Free Polyphenol Extract of Cherry on the Body Weight and DAI Index of UC Mice

There was no abnormal death during the whole period of study. On the fourth day of the administration, mice fed with DSS showed obvious symptoms, such as listlessness and hair getting rough (Figure 1(A2)), and we found that these mice had bloody stools and diarrhea. This indicated that the UC mice model induced by DSS might be established. Weight loss was a typical visible symptom of UC mice [40,43]. As shown in Figure 1B, the body weight of normal mice increased gradually; however, the body weight of mice orally treated with DSS showed apparent declines (*p* < 0.05). At the end of the experiment, the body weight of control mice was about 12% lower than at the first day of the study (decrease from 26 g to 23 g). Such losses were significantly alleviated by SASP or the free polyphenol extract of cherry, and there were no significant differences between the body weight of SASP and the extract (low and high doses)-managed mice (*p* < 0.05) (Figure 1C). The DAI is a comprehensive index based on weight loss, stool consistency and bloody stool score, and it is commonly used for evaluating the severity of UC; the higher the index, the more serious the disease [40]. In general, the changing tendency of the DAI index was similar to that of the body weight among the C, N, P, L and H groups, that is, the control mice exhibited the highest DAI index (3.15), and the extract-treated mice displayed a similar DAI index to that of the SASP treated mice (Figure 1D).

#### 3.4.2. Effect of the Free Polyphenol Extract of Cherry on Colon Length, and the Thickness of Colonic Mucous Layer of UC Mice

The visible picture of colon length, and the measured values of colon length and colonic thickness (mucous layer) were illustrated in Figure 2. The differences in the colon length among the five groups were remarkable (*p* < 0.01) (Figure 2A,B). Colon shortening was most severe in the C group (5.75 cm), which was about 30% lower than that of N mice (8.21 cm). The SASP or the cherry extract administration (low and high dose) significantly alleviated the DSS-induced colon shortening, and the three groups displayed similar colon lengths (*p* < 0.01, Figure 2A,B).

The colonic mucosal thickness of C mice were thicker than that of N mice, suggesting that DSS induced severe colonic ulcer and edema (Figure 2C). The cherry extract significantly reduced the thickness of colonic mucosa layer (low and high doses), which was similar to that of SASP-treated mice (*p* < 0.01).

#### 3.4.3. Effect of the Free Polyphenol Extract of Cherry on the Histological Properties of Colon of UC Mice

The pictures of the histological structure of the colons of the tested mice are shown in Figure 3, and the histological scores (Appendix A) were also performed to assess the degree of colitis of the tested mice; the higher the score, the more severe the injury. The histological structures of the colons of N mice were normal, that is, the structure of mucosa, submucosa, muscularis and serosal layer were intact, and the mucosal surface was covered with a single columnar epithelium, and the columnar epithelium cells were normal in shape (Figure 3A). However, C mice showed obvious colonic mucosal injuries, that is, the cytoplasm dissolved and formed an empty network with the depletion of goblet cell, as well as a large infiltration of inflammatory cell (mainly lymphocytes and neutrophils) (Figure 3B). Such a serious colon injury was also verified by the highest score of C mice (Appendix A). The management of SASP (Figure 3C) or the free polyphenol extract of cherry (Figure 3D,E) attenuated the injury of colonic mucosal, and the specific performance of the attenuation was that only a few inflammatory cells (mainly lymphocytes) were found around the lamina propria, and the goblet cells were slightly reduced. The histological structure of colonic tissues of P mice was similar to that of H mice, and the two groups also exhibited same histological scores (Appendix A).

#### 3.4.4. Effect of the Free Polyphenol Extract of Cherry on the Activity of Serum Enzymes, and the Levels of Inflammatory Cytokines of UC Mice

To further verify the effect of the free polyphenol extract of cherry on ulcerative colitis in mice, we measured the activity of enzymes (ALT, CAT, GSH-Px, SOD) (Figure 4), and the levels of inflammatory cytokines (MDA, MPO, NO) (Figure 5).

ALT was the marker of inflammation. C mice showed the highest level of serum ALT, while the damage was significantly ameliorated by the polyphenol extract (the high or low dose) or SASP (*p* < 0.01, Figure 4A).

CAT could break down hydrogen peroxide and decrease the level of hydroxyl radicals in vivo, and thus reduce oxidative damage to the body. The CAT activity of C mice was significantly lower than that of the N mice (*p* < 0.01), indicating that the DSS treatment induced oxidative damage in mice due to the production of hydroxyl radicals. There was no remarkable variation between the CAT activity of P mice or L mice and the CAT activity of C mice (*p* < 0.05). However, the high dose of the polyphenol extract clearly improved CAT activity (*p* < 0.05) (Figure 4B).

GSH-Px played an important role in protecting the structure and function of the cell membrane from oxidation damage by specifically catalyzing the reduction of hydrogen peroxide. Among all the tested mice, the C mice showed the lowest GSH-Px activity. SASP or the polyphenol extract treatment resulted in significant elevation in GSH-Px activity (*p* < 0.05) (Figure 4C).

The activity of SOD indirectly demonstrated the ability of the body to scavenge oxygen free radicals. The SOD activity of C mice was significantly lower than that of the N, P, or H mice (*p* < 0.01). This revealed that DSS reduced the ability to scavenge for oxygen free radicals in mice. The high-dose of the polyphenol extract significantly increased the activity of SOD to nearly that of the SASP management. However, compared to DSS, the low dose of the polyphenol extract did not clearly improve SOD activity (*p* < 0.05) (Figure 4D).

The content of MDA indicated the degree of lipid peroxidation and cell damage. C mice showed a marked increase in MDA content (*p* < 0.01), indicating that DSS aggravated lipid peroxidation and cell damage in mice. The polyphenol extract (high or low doses) significantly reduced the MDA level (*p* < 0.05), and a much more remarkable decrease in MDA content was introduced by the administration of SASP (*p* < 0.01) (Figure 5A).

MPO is a pro-oxidation and pro-inflammatory enzyme, and its activity is regarded as an important inflammatory indicator of ulcerative colitis. The activity of colonic MPO of C mice was about 2.5-fold higher than that of N mice, demonstrating that DSS treatment induced serious ulcerative colitis. The administration of SASP or the polyphenol extract significantly decreased the activity of MPO (*p* < 0.05), suggesting their potential protective effects on DSS-induced ulcerative colitis in vivo (Figure 5B).

NO is a reactive free radical in vivo, and the excessive production of NO was closely associated with inflammatory diseases [44]. DSS caused a significant increase in NO content in colon (*p* < 0.05). Although SASP or the polyphenol extract treatment decreased the NO level, such declines were not statistically significant (*p* < 0.05) (Figure 5C).

#### 3.4.5. Effect of the Free Polyphenol Extract of Cherry on Levels of Inflammatory Factors of UC Mice

IL-6 was a pro-inflammatory cytokine that inhibited the development of the regulatory T cells [45]. TNF-α was also a pro-inflammatory factor, and the inhibitor of TNF-α was one of the pharmacological agents used for treating inflammatory bowel disease [44]. The levels of IL-6 (Figure 6A) and TNF-α (Figure 6B) of C mice were the highest among all the tested mice, suggesting that the administration of DSS prompted the production of pro-inflammatory factors in mice. However, the management of the polyphenol extract remarkably decreased the quantities of IL-6 and TNF-α, and such declines induced by the high dose of the polyphenol extract were even clearer (*p* < 0.01).

IL-10 was an important immunoregulatory cytokine, and mainly produced by T cells [45]. Compared to N mice, C mice exhibited lower levels of IL-10. On the contrary, the free polyphenol extract of cherry (especially the high dose) significantly increased the IL-10 level (*p* < 0.05, Figure 6C).

It suggested that the free polyphenol extract of cherry likely acted as the inhibitor of IL-6 and TNF-α but the promotor of IL-10, and thus contributed to relieve ulcerative colitis induced by DSS in a cytokine-specific manner.

### 3.5. The Possible Mechanisms of the Free Polyphenol Extract of Cherry on DSS-Induced Ulcerative Colitis in Mice

#### 3.5.1. Effect of the Free Polyphenol Extract of Cherry on Tight Junction (TJ) Proteins

The staining photographs of immunofluorescence analysis of ZO-1 and occludin proteins are shown in Figure 7, and the quantities of ZO-1 and occludin are exhibited in Figure 8.

As shown in Figure 7, the substrate was white, the negative cells were blue, and the positive cells were yellow or claybank. The positive products of occludin and ZO-1 mainly distributed in the cytoplasm and intercellular stroma. Generally, the positive-stained areas of occludin and ZO-1 of colonic mucosa only treated by DSS were smaller than those of the colonic mucosa, also managed by SASP or the polyphenol extract.

The results of the quantitative analysis declared that the levels of occludin and ZO-1 in C mice were 4.8% and 6.0% lower than those in N mice, and such declines were significant (*p* < 0.05, Figure 8). Compared to the management of single DSS, the combined treatment with the polyphenol extract remarkably elevated the content of ZO-1 by 4.7%; the increase was 4.3% for occludin. The SASP administration also increased the levels of occludin and ZO-1 by 4.6% and 3.9%, respectively. It demonstrated that the free polyphenol extract of cherry had the potential to prompt the intestinal barrier function.

#### 3.5.2. The Free Polyphenol Extract of Cherry Inhibited the Wnt/β-Catenin Pathway

The Wnt pathway played an essential role in maintaining homeostasis in intestinal epithelial cells, and it is commonly regarded as a driving factor for the repair of colonic injury [46,47]. One of the most typical Wnt signaling pathways is the Wnt/β-catenin pathway, the role of which depends on the intracellular level of β-catenin; the abnormally increased level of intracellular β-catenin was involved in the development of ulcerative colitis [48]. We measured the expression levels of β-catenin, c-myc, cyclin D1 and GSK-3β, which were involved in the Wnt/β-catenin signaling pathway, in order to assess whether the oral pretreatment with cherry polyphenol weakened the abnormal activation of Wnt/β-catenin signaling pathway.

As shown in Figure 9, compared to N mice, C mice exhibited clearly higher expression levels of β-catenin (1.4-fold higher), c-myc (1.9-fold higher), cyclin D1 (3-fold higher) and GSK-3β (3.4-fold higher) (*p* < 0.01), suggesting that DSS administration induced the abnormal activation of Wnt/β-catenin pathway. However, the treatment by the polyphenol extract remarkably downregulated the expression of the four proteins, and such decreases were in a dose-dependent manner. Especially, the high dose (450 mg/kg BW) reduced the expression levels of β-catenin, c-myc, cyclin D1 and GSK-3β by approximately 43%, 35%, 50% and 48%, respectively. This indicated that the free polyphenol extract of cherry weakened the abnormal activation of the Wnt/β-catenin signaling pathway, and thus contributed to relieve ulcerative colitis in mice.

## 4. Discussion

The reduction in ulcerative colitis symptoms was the aim of the treatment for ulcerative colitis. DAI was calculated from the disease signs and symptoms, and it was commonly regarded as a standard indicator for judging the degree and prognosis of UC [49,50]. We examined the effect of the 450 and 150 mg/kg cherry polyphenol extract on DSS-induced ulcerative colitis in mice. The results suggested that both doses of the extract significantly decreased the DAI index, which was clearly increased by a single DSS (*p* < 0.01, Figure 1D). The alleviating effect induced by the cherry extract on UC was also verified by some visual symptoms, such as slower weight loss (Figure 1B,C), reduced shortening of colon length (Figure 2A,B), and decreased thickness of the colonic mucosal layer (Figure 2C). The positive effect of the cherry extract in alleviating symptoms of ulcerative colitis might be the main contribution to polyphenols. It reported that the dietary administration of fruits enriched in polyphenols could significantly reduce weight loss, while increasing the colon length in mice with ulcerative colitis [31,37].

One of the important pathogenic factors associated with ulcerative colitis is oxidative stress, and the inflammatory cytokine was an important transcription factor that regulated the inflammation process in ulcerative colitis [51]. The origin of reactive oxygen species in inflamed mucosa mainly consisted of superoxides and peroxides, which eventually led to oxidative stress or further oxidative damage [50]. CAT, GSH-Px and SOD could reduce oxidative stress by decomposing peroxides and superoxides. Therefore, we further evaluated the positive effectiveness of the cherry extract by measuring the antioxidant activity, including ALT, CAT, GSH-Px, SOD and MDA in serum, as well as the inflammatory cytokine levels (MPO, NO) of colon tissue. Our study showed a significant elevation in ALT, MDA, MPO and NO levels (Figure 4 and Figure 5) and a significant decline in SOD, CAT and GSH-Px (Figure 4) in the C group in comparison with the normal group. This indicated that the lipid peroxidation and oxidative stress was in the colon tissue [52]. Mice treated with the cherry extract (150 and 450 mg/kg) showed a significant reduction in MDA, MPO and NO levels and a significant elevation in SOD, CAT and GSH-Px levels, and such positive influence was recorded in a dose-depended manner (Figure 4 and Figure 5). As illustrated in Table 1, the cherry extract was composed of anthocyanins, hydroxycinnamic acids, flavonols, etc. The identified polyphenols all have great antioxidant capacities due to their phenolic hydroxyl structures, which ensure that the polyphenols have the ability to scavenge the peroxyl radicals and interrupt a peroxidative chain reaction, and thus they had a positive influence on UC [53].

The typical feature of ulcerative colitis was the inflammation of the intestinal mucosa, which resulted from the activation of the immune response by pro-inflammatory mediators (IL-6 and TNF-α) and anti-inflammatory cytokine IL-10 [54,55]. TNF-α was a primary inflammatory cytokine and stimulated immune cells to produce and release other cytokines, chemokines, reactive oxygen and nitrogen species, etc. [56]. TNF-α played a fundamental role in the pathogenesis of UC, and the anti-TNF-α agents were commonly used for UC treatment [57]. IL-6 induced the expression of acute-phase proteins during acute inflammation, and the levels of TNF-α and IL-6 in serum and colon tissue closely correlate with the severity of intestinal inflammation [58]. IL-10 attenuated UC in vivo through reducing oxidative stress, and inhibited the synthesis of pro-inflammatory cytokines (TNF-α) [51,59]. We found that the administration of the cherry extract caused a significant decrease in the colonic levels of IL-6 and TNF-α (*p* < 0.01), but a remarkable elevation in the content of IL-10 compared to treatment by a single DSS (*p* < 0.05, Figure 6). These results indicated that the cherry extract had a remarkable anti-inflammatory property by regulating the balance between anti-inflammatory and pro-inflammatory cytokines.

The intestinal epithelium played a fundamental role in separating internal tissues, maintaining homeostasis, fixing intestinal morphology and regulating intestinal function [60]. The tight junction (TJ) proteins were responsible for connecting colonocytes and maintaining the integrity of epithelium and the homeostasis of intestinal barrier. Disturbances of TJ were involved in the pathogenesis of inflammatory bowel diseases [15]. The ZO and occludin were the typical TJ proteins [15,61]. The present study found that the cherry extract significantly increased the levels of ZO, occluding to the single DSS levels (Figure 8). This suggested that the free polyphenol extract of cherry had the potential to enhance the colon barrier, and thus ameliorate UC induced by DSS in mice.

The Wnt signaling pathway was involved in the proliferation and differentiation of intestinal stem cells, and therefore responsible for the renewal and replacement of colonic epithelial cells. The function of the Wnt signaling pathway depended on the level of β-catenin, which was regulated by the phosphorylation levels of protein complexes and their degradation [62]. The level of β-catenin was closely associated with the progress of UC. When colonic mucosa was damaged, β-catenin accumulated and moves to the nucleus, which activated the transcription of cell-proliferation-related genes (c-myc and cyclin D1), aggravating UC [63]. It might be the reason why the C mice exhibited higher levels of β-catenin, c-myc and cyclin D1 than the N mice (Figure 9).

The most typical Wnt signaling pathway was the Wnt/β-catenin signaling pathway, an evolutionarily conserved signal transduction pathway [64]. When the active Wnt ligand was absent, β-catenin would bind to Axin (the scaffold proteins axis inhibition protein) and APC (adenomatosis polyposis coli), and interact with GSK-3β leading to the phosphorylation of four n-terminal residues of β-catenin (Figure 10). The phosphorylated β-catenin controlled by APC, GSK-3β, Axin2, and CK1 (casein kinase 1) was a target for ubiquitination and proteosomal degradation [65]. When the Wnt ligand was present, β-catenin would bind to membrane receptor frizzled (FZD) and the low-density lipoprotein receptor-related protein (LRP), leading to the cytoplasmic LRP phosphorylation by GSK-3β, which further caused the aggregation of cytoplasmic proteins, Dishevelled (Dvl) and Axin, and then the β-catenin phosphorylation and protein degradation were inhibited [64]. The accumulated β-catenin was transferred to the nucleus, and then the regulated c-myc, cyclin D1, metalloproteinase-7 and peroxisome proliferation-activated receptor-d were transferred through TCF (T cell factor) and LEF (lymphoid enhancer factor) transcription complex [62] (Figure 10).

The Wnt/β-catenin signaling pathway is essential for the generation and maintenance of intestinal crypt structure, and it is one of the major pathways in intestinal crypt proliferation and differentiation, as well as the development of UC [47,65].

Nuclear translocation of β–catenin was induced in UC; however, its mechanism of action is still unclear. Previous studies declared that the 14-3-3ζ protein [66] and PI3K regulatory subunit p85α [67,68] in the cytoplasm might play a role in regulating the nuclear translocation of β-catenin and the transcriptional activation of TCF/LEF. The movement of β-catenin to the nucleus appeared, even prior to the infiltration of neutrophil [69]. The β-catenin accelerated UC by inducing the proinflammatory properties of T cells [48]; GSK-3β/β-catenin was related to other mediators (p38 MAPK and PPARγ). Polyphenols may alleviate the increase in β-catenin by reducing the levels of PPARγ and p38 MAPK to relieve UC [48]. Therefore, as shown in Figure 9, the inhibited expression of β-catenin, c-myc, Cyclin D1 and GSK-3β by the free polyphenol extract of cherry (especially the high dose) suggested its great potential against UC.

## 5. Conclusions

This study showed that the free polyphenol extract of cherry exerted protective effects against DSS-induced ulcerative colitis. The results of UPLC-ESI-TOF-MS/MS analysis showed that the main compounds in the free polyphenol extract of cherry were anthocyanins. Therefore, this report suggested potential beneficial effects of cherry anthocyanins on ulcerative colitis. Indeed, the free polyphenol extract of cherry elevated levels of TJ proteins (ZO-1 and occluding) to prompt the intestinal barrier function. Otherwise, the free polyphenol extract of cherry might regulate oxidative stress by enhancing CAT, GSH-Px and SOD capacities of serum and decreasing inflammatory cytokine levels (MPO, NO) of colon tissue. The free polyphenol extract of cherry also weakened colonic inflammation associated with oxidative stress by decreasing the levels of pro-inflammatory mediators (IL-6 and TNF-α) and elevating levels of anti-inflammatory cytokine IL-10. Further, we found that the free polyphenol extract of cherry prevented the upregulation of various pro-inflammatory genes (β-catenin, c-myc, cyclin D1 and GSK-3β). It was, therefore, inferred that the free polyphenol extract of cherry alleviated UC by suppressing the Wnt/β-catenin signaling pathway.

## Figures and Tables

**Figure 1 foods-11-00049-f001:**
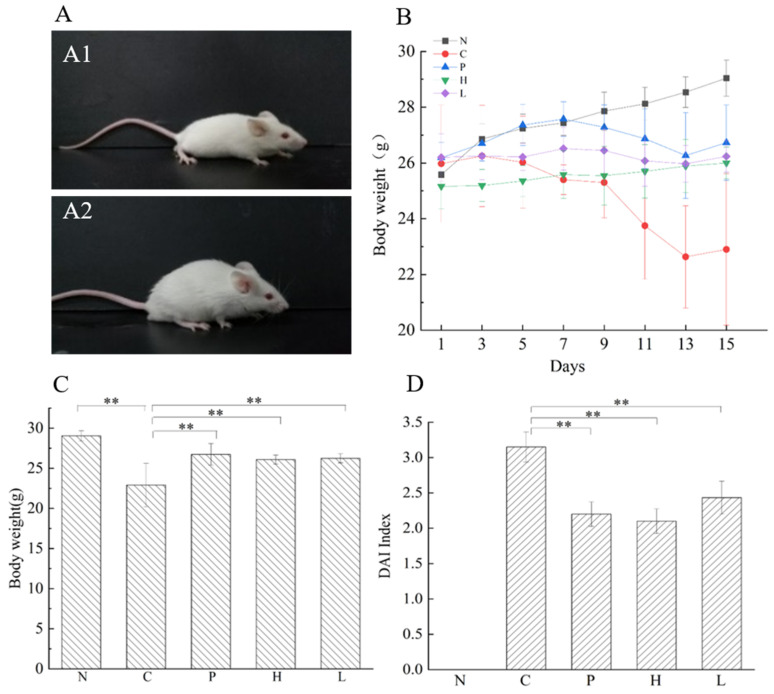
(**A**) Effect of the free polyphenol extract of cherry on the appearance, ((**A1**) before setting up the UC model, (**A2**) after setting up the UC model), DAI index (**B**), body weight change over time (**C**) and body weight at the end of the experiment (**D**), of the tested mice. The ** symbol indicated that differences among the compared groups were significant at *p* < 0.01 level.

**Figure 2 foods-11-00049-f002:**
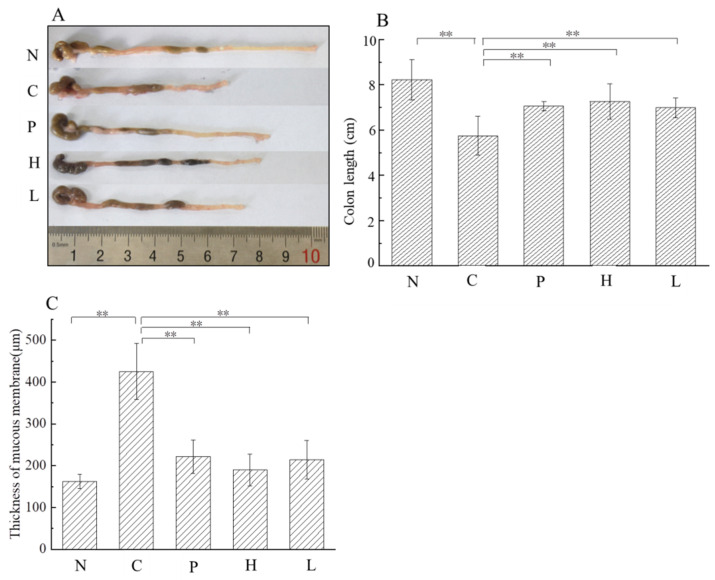
Effect of the free polyphenol extract of cherry on colon length (**A**,**B**) and thickness of mucous membrane (**C**) of the tested mice. ** indicated that differences were significant at *p* < 0.01 level.

**Figure 3 foods-11-00049-f003:**
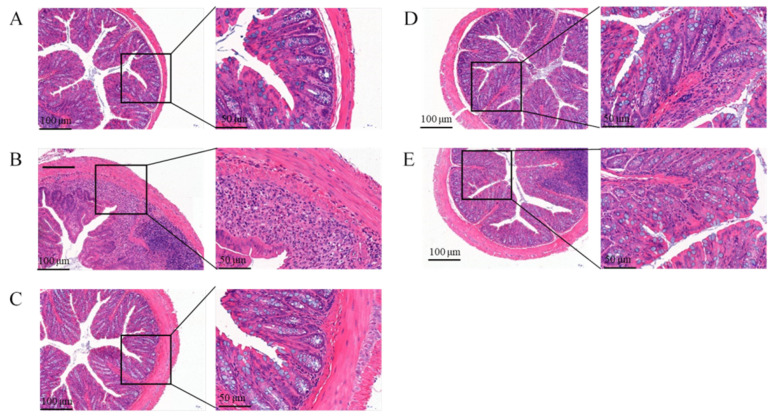
The colonic histopathology of the tested mice (100×, 400×). (**A**). The colon of normal mice; (**B**). The colon of control mice; (**C**). The colon of positive mice; (**D**). The colon of mice treated by the high dose (450 mg/kg BW) of the free polyphenol extract of cherry; (**E**). The colon of mice treated by the low dose (150 mg/kg BW) of the free polyphenol extract of cherry.

**Figure 4 foods-11-00049-f004:**
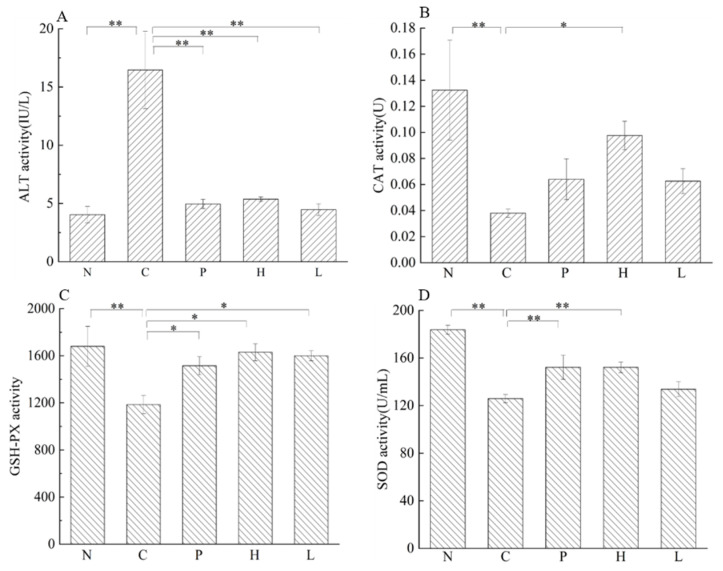
Effect of the free polyphenol extract of cherry on activity of ALT (**A**), CAT (**B**), GSH-Px (**C**) and SOD (**D**) in serum of the tested mice. ** indicated that differences among the compared groups were significant at *p* < 0.01 level, and * showed that differences were significant at *p* < 0.05 level.

**Figure 5 foods-11-00049-f005:**
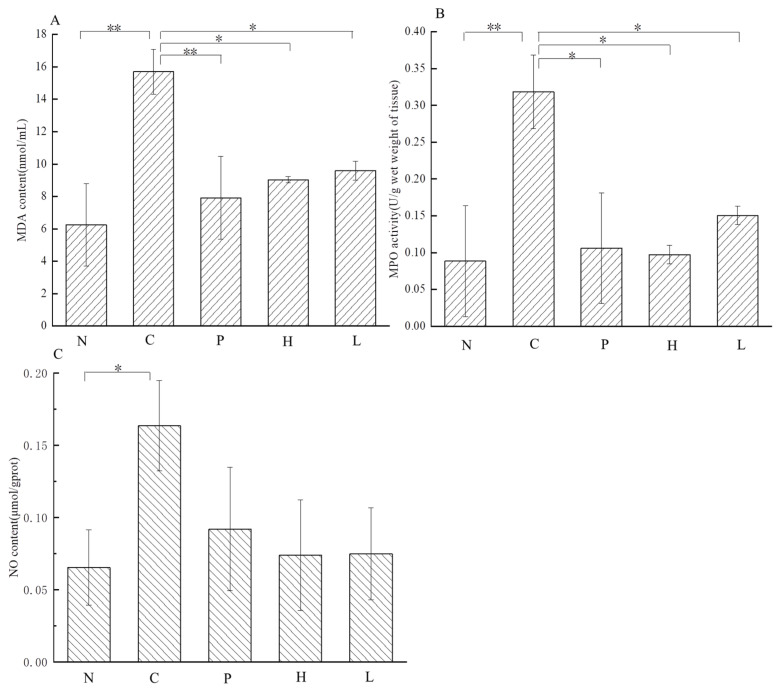
Effect of the free polyphenol extract of cherry on levels of serum MDA (**A**), and MPO (**B**) and NO (**C**) in colonic tissue of the tested mice. ** indicated that differences among the compared groups were significant at *p* < 0.01 level, and * showed that differences were significant at *p* < 0.05 level.

**Figure 6 foods-11-00049-f006:**
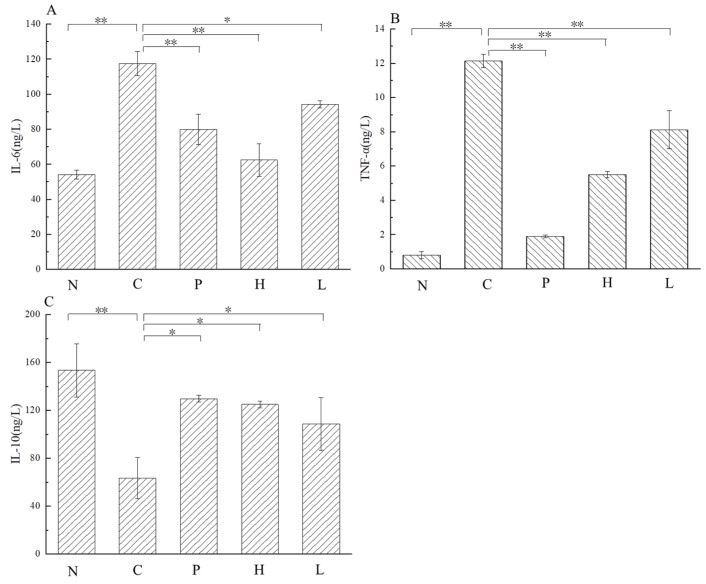
Effect of the free polyphenol extract of cherry on levels of IL-6 (**A**), TNF-α (**B**) and IL-10 (**C**). ** indicated that differences among the compared groups were significant at *p* < 0.01 level, and * showed that differences were significant at *p* < 0.05 level.

**Figure 7 foods-11-00049-f007:**
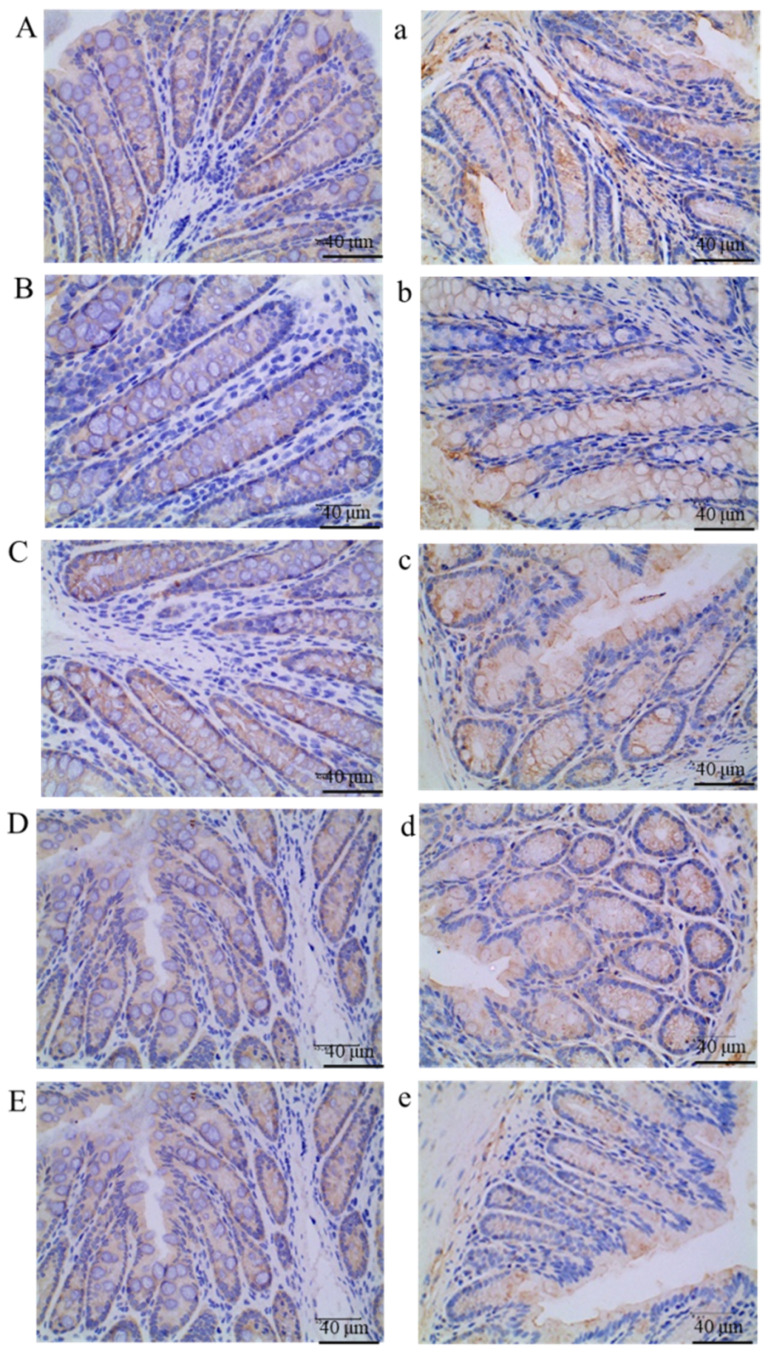
The staining photographs of immunofluorescence analysis of ZO-1 (**A**–**E**) and occludin (**a**–**e**) proteins in colonic tissue of mice (400×). A/a. N mice; B/b. C mice; C/c. P mice; D/d. H mice; E/e. L mice.

**Figure 8 foods-11-00049-f008:**
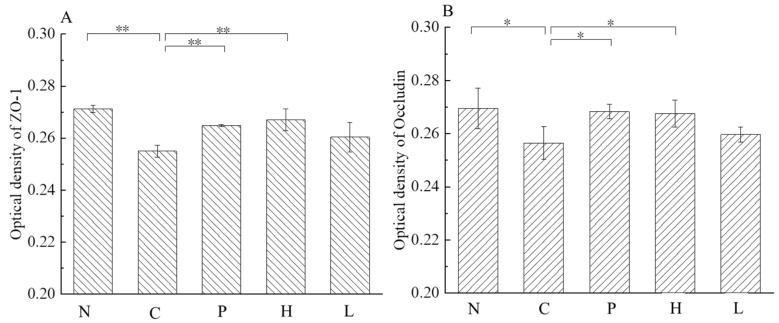
Effect of the free polyphenol extract of cherry on quantities of ZO-1 (**A**) and occluding (**B**). ** indicated that differences among the compared groups were significant at *p* < 0.01 level, and * showed that differences were significant at *p* < 0.05 level.

**Figure 9 foods-11-00049-f009:**
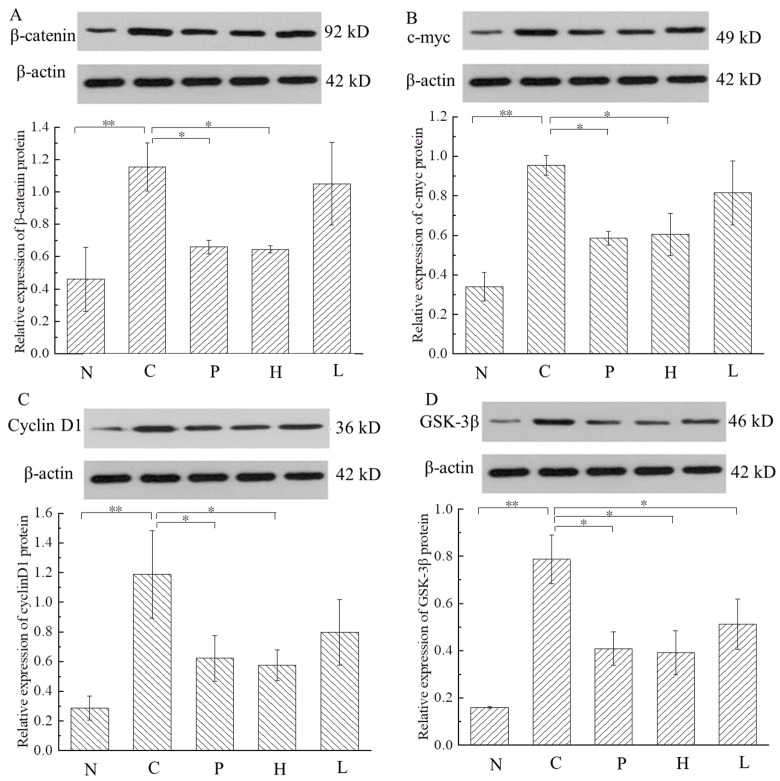
Effect of the free polyphenol extract of cherry on the expression of β-catenin (**A**), c-myc (**B**), Cyclin D1 (**C**) and GSK-3β (**D**) in colonic tissue of the tested mice. The ** symbol indicated that differences among the compared groups were significant at *p* < 0.01 level, and * showed differences were significant at *p* < 0.05 level.

**Figure 10 foods-11-00049-f010:**
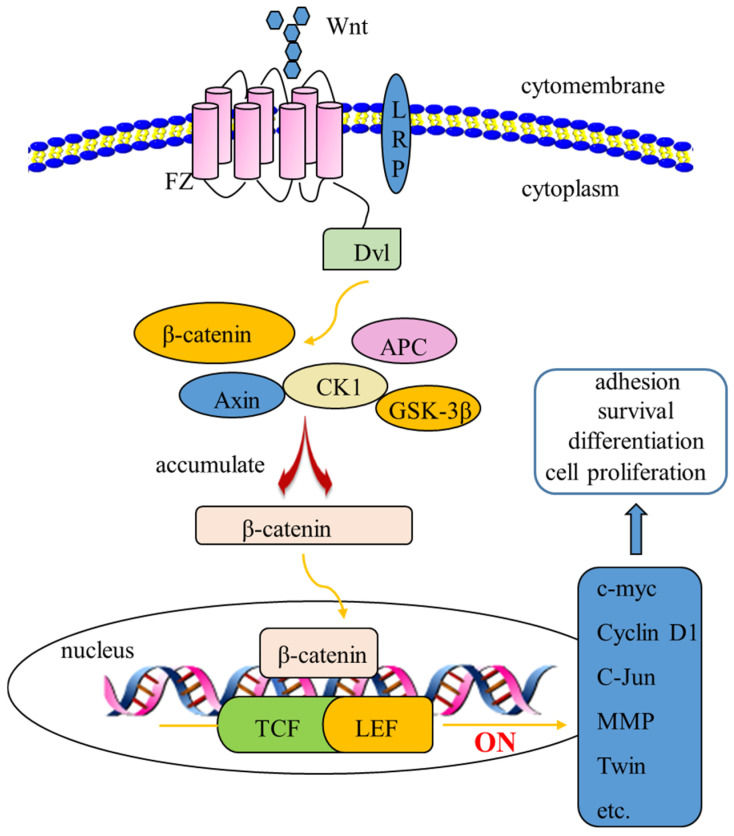
The simplified representation of Wnt/β-catenin signaling pathway.

**Table 1 foods-11-00049-t001:** Identification of the free and bound polyphenol of the cherry extract by UPLC-ESI-TOF-MS/MS.

	Compounds	T_R_ ^a^ (min)	Molecular Formula	(M + H)^+^	MS^2^ ion Fragments (*m*/*z*)	Error (ppm)	Ref
**Free anthocyanins**							
1	Cyanidin 3-*O*-glucoside	1.169	C_21_H_20_O_11_	449.1103	287.0569 (100%)	3.4	[30]
2	Cyanidin 3-*O*-rutinoside	1.014	C_27_H_30_O_15_	595.1691	287.0561 (100%)	5.6	[30]
3	Peonidin 3-*O*-rutinoside	1.288	C_28_H_33_O_15_	610.1888	302.0759 (100%), 301.0759 (87%)	−0.6	[30]
4	Malvidin-hexoside	3.659	C_23_H_25_O_12_	493.1361	331.0817 (100%)	3.1	[41]
5	Pelargonidin 3-rutinoside	1.588	C_27_H_30_O_14_	579.1537	271.0613 (100%), 127.0389 (79%), 139.0388 (45%)	6.9	[41]
**Bound anthocyanins**							
6	Delphinidin 3-*O*-rutinoside	1.648	C_27_H_30_O_16_	611.1632	303.0498 (100%)	3.2	[30]
7	Delphinidin-hexoside	1.920	C_21_H_20_O_12_	465.1038	303.0514 (100%)	2.3	[30]
**Free hydroxybenzoic acid**							
8	Protocatechuic acid	3.096	C_7_H_6_O_4_	155.0348	68.9973 (100%), 109.0284 (13%)	2.3	[30]
**Free hydroxycinnamic acids**							
9	Caffeoylquinic acid	1.502	C_16_H_18_O_9_	355.1048	163.0398 (100%), 145.0289 (28%), 117.0337 (15%)	1.7	[30]
10	3-*p*-Coumarylquinic acid	1.584	C_16_H_18_O_8_	339.1083	147.0443 (100%), 119.0488 (30%)	2.7	[30]
11	Feruloylquinic acid	2.111	C_17_H_20_O_9_	369.1193	145.0292 (100%), 177.0549 (91%)	3.5	[41]
12	*di*-Caffeoylquinic acid	2.731	C_25_H_24_O_12_	517.1368	163.0394 (100%)	5.3	[41]
**Bound hydroxycinnamic acid**							
13	*p*-Coumaric acid	2.698	C_9_H_8_O_3_	165.0551	91.0538 (100%), 119.0485 (57%)	2.7	[41,42]
**Free flavonols**							
14	Quercetin 3-*O*-rutinoside	1.698	C_27_H_30_O_16_	611.1648	303.0518 (100%)	4.1	[30]
15	Quercetin	1.744	C_15_H_10_O_7_	303.0510	303.0514 (100%) 257.0450 (15%), 229.0497 (33%), 153.0180 (40%)	3.6	[30]
16	Quercetin 3-*O*-hexoside	1.741	C_21_H_20_O_12_	465.1048	303.0509 (100%)	3.1	[30]
17	Quercetin-7-*O*-glucoside-3-*O*-rutinoside	1.551	C_33_H_41_O_21_	773.2135	303.0519 (100%), 465.0962 (65%)	−0.7	[30]
**Bound flavonol**							
18	Kaempferol 3-glucoside	2.441	C_21_H_20_O_11_	449.1099	287.0570 (100%)	4.6	[21]
**Free flavanols**							
19	Catechin	1.755	C_15_H_14_O_6_	291.0872	139.0388 (100%), 123.0435 (80%), 147.0439 (32%)	3.1	[41]

^a^ The time appeared in the secondary mass spectra of the identified compounds.

## Data Availability

This study did not report any data.

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
