# Peer review of "Cherry Polyphenol Extract Ameliorated Dextran Sodium Sulfate-Induced Ulcerative Colitis in Mice by Suppressing Wnt/β-Catenin Signaling Pathway"

_foods, 2021, doi:10.3390/foods11010049_

Round 1
Reviewer 1 Report
The presented study aimed to investigate the potential role of the cherry polyphenol extract in alleviating ulcerative colitis and its possible mechanism of action. This work indicates that the cherry polyphenol extract treatment is able to attenuate DSS-induced colitis in mice by inhibition of inflammatory and oxidative stress mediators through Wnt/β-catenin signaling pathway. The findings revealed that the cherry polyphenol-rich extract management significantly alleviated ulcerative colitis symptoms, such as weight loss, colon shortening, thickening of colonic mucous and muscular layer, etc.
The manuscript is clearly presented and the subject is suitable for Foods , but some points should be revised or supplemented.
- The first and most important objection: no quantitative chromatographic analysis of phenolic compounds have been performed during experiment.
- In my opinion, Authors should supplement the research with simple analyzes of the composition and properties of the tested extract, e.g. total polyphenol content, DPPH/ABTS free radical scavenging activity...
- The section 2.1. Materials is incomplete. The Authors did not provide information on the vouchers of the plants.
- The section 2.3. Identification of cherry polyphenol extract by UPLC-ESI-TOF-MS/MS should be described in more detail (injection volume etc.).
- Conclusions are incomplete and should be supplemented.
Author Response
To Reviewer 1:
The presented study aimed to investigate the potential role of the cherry polyphenol extract in alleviating ulcerative colitis and its possible mechanism of action. This work indicates that the cherry polyphenol extract treatment is able to attenuate DSS-induced colitis in mice by inhibition of inflammatory and oxidative stress mediators through Wnt/β-catenin signaling pathway. The findings revealed that the cherry polyphenol-rich extract management significantly alleviated ulcerative colitis symptoms, such as weight loss, colon shortening, thickening of colonic mucous and muscular layer, etc.
The manuscript is clearly presented and the subject is suitable for Foods , but some points should be revised or supplemented.
- The first and most important objection: no quantitative chromatographic analysis of phenolic compounds have been performed during experiment.
√ The quantitative chromatographic analysis of phenolic compounds was performed by HPLC, and the results were added in part 3.2.
- In my opinion, Authors should supplement the research with simple analyzes of the composition and properties of the tested extract, e.g. total polyphenol content, DPPH/ABTS free radical scavenging activity...
√ The composition of the tested extract was analyzed by HPLC, the total content of the cherry extract tested by the Folin-Ciocalteu method was supplemented. We also estimated the antioxidant capacities of the free and bound polyphenols extract by using ABTS, DPPH, FRAP, and ORAC assays. Results of the total polyphenol content and antioxidant capacities were illustrated in Section 3.1 in the revised manuscript.
- The section 2.1. Materials is incomplete. The Authors did not provide information on the vouchers of the plants.
√ More information on the tested cherry was supplemented in lines 72-75.
The section 2.3. Identification of cherry polyphenol extract by UPLC-ESI-TOF-MS/MS should be described in more detail (injection volume etc.). P
√ The details about the injection volume and other MSn detection parameters were added, Please check it in Part 2.4 in the revised manuscript.
- Conclusions are incomplete and should be supplemented.
√ It has been enriched.

Reviewer 2 Report
The submitted work is interesting, deals with a relevant topic and is well presented, however some points need to be addressed. Please find below my comments and suggestions.
- It is not recommended to use abbreviations in the title of the manuscript;
- It is not clear what species of cherry is discussed. The full botanical name of the cherry is required;
- Why was “Meizao” variety chosen?
- Lines 88-89. There are no references to the Folin-Ciocalteu method;
- Section 2.3. ESI conditions (temperature levels of ESI interface, desolvation line, heat block, nebulizing gas, heating gas, collision energy, scanning range etc.) must be specified;
- Why did the authors use a preparative column (100×21 mm)?
- Three separate groups of mice treated only with cherry polyphenol extracts (450 and 150 mg/kg) and SASP (300 mg/kg) should be included in order to assess their influence on mice;
- Table 1. Renumber column 1;
- Table 1. The data in column 5 does not refer to [M]+, but to [M+H]+;
- Why did quercetin and quercetin-3-O-rutinoside have the same tr (1.75)?
- Where is Section 3.2.2?
- Sections 3.2.1., 3.2.3., 3.2.4 etc. Specify the name of the polyphenol extract in subtitles;
- Figure 1, 2, 4 etc. Where is the explanation of *?
- According to Figure 2, the results obtained for D (muscular thickness) are irrelevant;
- Lines 367-368. According to Section 3.1, polyphenols accounted for only 0.68% of the weight of cherry extract. Thus, 99% of the extract remains uncharacterized. It is incorrect to say that “polyphenols positive effect of the cherry extract in alleviating symptoms of ulcerative colitis might be mainly contributed to polyphenols.”
- Conclusion is too short.

Author Response
Manuscript ID: foods-1476793
Title: Cherry polyphenol extract ameliorated dextran sodium sulfate-induced ulcerative colitis in mice by suppressing Wnt/β-catenin signaling pathway
Dear Professors,
Thank you for your useful comments and suggestions on our manuscript. We have modified the manuscript accordingly, and all changes were marked in red in the text, and the detailed response to reviewers were listed as follows:
To Reviewer 2:
The submitted work is interesting, deals with a relevant topic and is well presented, however some points need to be addressed. Please find below my comments and suggestions.
- It is not recommended to use abbreviations in the title of the manuscript;
√ The abbreviation DSS was written in full, and the title was revised as “Cherry polyphenol extract ameliorated dextran sodium sulfate -induced ulcerative colitis in mice by suppressing Wnt/β-catenin signaling pathway”.
- It is not clear what species of cherry is discussed. The full botanical name of the cherry is required;
√ Both the breed name and its origin of hybridization was illustrated in Section 2.1 Materials.
- Why was “Meizao” variety chosen?
√ Two reasons for the chosen.
(1)Meizao cherry has advantages of large fruit, high quality, good storage and transportation, common on the market and popular with the public.
(2)According to the recent review of cherry (Blando F, Oomah B D. Sweet and sour cherries: Origin, distribution, nutritional composition and health benefits[J]. Trends in Food Science and Technology, 2019, 86: 517-529.), no report has focused on the positive effect of Meizao cherry on ulcerative colitis in mice, therefore, the Meizao cherry was chosen.
- Lines 88-89. There are no references to the Folin-Ciocalteu method;
√ The reference to the Folin-Ciocalteu method was added.
- Section 2.3. ESI conditions (temperature levels of ESI interface, desolvation line, heat block, nebulizing gas, heating gas, collision energy, scanning range etc.) must be specified;
√ These information has been supplemented in Part 2.4.
- Why did the authors use a preparative column (100×21 mm)?
√ It is not a preparative column (100×21 mm), but a separation column (150×4.6 mm), we have corrected it, and thank you for pointing out our mistake.
- Three separate groups of mice treated only with cherry polyphenol extracts (450 and 150 mg/kg) and SASP (300 mg/kg) should be included in order to assess their influence on mice;
√ From the point of view of academic rigour and comprehensiveness, we totally agree with the reviewer`s suggestion, which has instructive significance for our future animal experiments. There were reasons why the three separate groups of mice treated only with cherry polyphenol extracts (450 and 150 mg/kg) and SASP (300 mg/kg) were not included in the original scheme of the animal experiment:
Generally, the current researches suggested that the polyphenols extract from plant materials has positive influence on ulcerative colitis in mice. Therefore, it could be inferred that the cherry polyphenol extract might not cause or aggravate ulcerative colitis in mice at least.
Otherwise, before the setting of the ulcerative colitis model, a 2-days administration was performed in advance, that is, the three separate groups of mice were only treated with cherry polyphenol extracts (450 and 150 mg/kg) and SASP (300 mg/kg) for two days (once daily), and the cherry extract or SASP treated mice did not showed any abnormal performance, such as weakness, lethargic and sluggish, dull fur and so on. Therefore, it was confirmed that the influence of cherry polyphenol extract and SASP was not negative at least.
Moreover, the SASP dose was selected based on the previous reports, and the doses of cherry extracts were calculated based on the daily intake of polyphenols in human diets by using body surface area method according to the literatures, and the experimental mice showed no death and no abnormal performance during the whole experiment. It could be concluded that the doses of the cherry polyphenol extracts (450 and 150 mg/kg) and SASP (300 mg/kg) were tolerable in mice.
Above all, it should be emphasized that this study mainly focused on the influence of cherry polyphenol extract on mice with ulcerative colitis. We attempted to explore whether the cherry polyphenol extract had the potential to relieve the symptoms of ulcerative colitis in mice and the possible mechanism of action instead of studying its influence on mice. We are truly appreciated the reviewer`s suggestion, which gave us a new way of thinking about the effect of edible compounds on health. Once again, thanks for your suggestion!
- Table 1. Renumber column 1;
√ They were numbered in order in Table 1.
- Table 1. The data in column 5 does not refer to [M]+, but to [M+H]+;
√ It was corrected accordingly.
- Why did quercetin and quercetin-3-O-rutinoside have the same tr (1.75)?
√ The tr presented in Table 1 of the first manuscript refers to the time that the peak appeared in base peak chromatogram (first order chromatogram). Generally, the tr was automatically derived by the mass spectrometry analysis system, and the decimal portion of the tr was rounded. The fact is that the third digit of the decimal of quercetin tr is different from that of quercetin-3-O-rutinoside tr . For example, quercetin tr might be 1.752 min, and quercetin-3-O-rutinoside tr might be 1.754 min, when rounded, the two compounds have the same tr (1.75).
It's worth noting that the tr appeared in secondary mass spectra of the identified compounds was unique and distinguishable. Therefore, we supplemented the tr appeared in secondary mass spectra of the identified compounds for clearly understanding.
- Where is Section 3.2.2?
√ The original Section 3.2.2 was revised to Section 3.4.2 in the modified manuscript.
- Sections 3.2.1., 3.2.3., 3.2.4 etc. Specify the name of the polyphenol extract in subtitles;
√ It was specified as the cherry polyphenol extract in subtitles.
- Figure 1, 2, 4 etc. Where is the explanation of *?
√ The explanation of * has been supplemented in the figure subtitles.
- According to Figure 2, the results obtained for D (muscular thickness) are irrelevant;
√ Here, the muscular thickness is the thickness of colonic muscle. Both the thickness of colonic muscle and mucous membrane layer were closely related to the pathology of the colon. We have revised the expression in the manuscript for clearly understanding.
- Lines 367-368. According to Section 3.1, polyphenols accounted for only 0.68% of the weight of cherry extract. Thus, 99% of the extract remains uncharacterized. It is incorrect to say that “polyphenols positive effect of the cherry extract in alleviating symptoms of ulcerative colitis might be mainly contributed to polyphenols.”
√ The total polyphenol content (TPC) of the cherry extract was about 6.84 mg GAE/g DW, here, the unit of TPC means mg of gallic acid equivalent per gram of edible part of dry cherry (mg GAE/g DW) instead of per gram of the extract. Therefore, it might be not suitable to think that 99% of the extract remains uncharacterized.
We used both HPLC and UPLC-ESI-TOF-MS/MS methods to explore the polyphenols composition and their content. A total of 15 polyphenols were identified and cyanidin 3-O-glucoside was the main phenolic compound in the free polyphenol extract of cherry. Otherwise, the cherry polyphenol extract was purified by AB-8 macroporous resin to remove impurities to ensure that the cherry polyphenol extract used in animal experiments was mainly consisted of polyphenols.
- Conclusion is too short.
√ It has been enriched.

Reviewer 3 Report
This manuscript describes the protective effects of cherry polyphenol extract on Dextran sodium sulfate-induced ulcerative colitis in mice. The manuscript needs some revisions to do, as following:
- The authors should use the abbreviations only after having written them in full. See: MPO in abstact. Moreover, in line 60 the authors used the abbreviation DSS, but only in the line 67 they explained what DSS stands for. In Line 106, SPF, please write specific-pathogen-free
- The authors could change the title of their manuscript, avoiding using the abbreviation DSS but writing it in full, like this “Cherry polyphenol extract ameliorated Dextran sodium sulfate -induced ulcerative colitis in mice by suppressing Wnt/β-catenin signaling pathway”. This is a suggestion that could make the title clearer also for all readers, even those not experts in this field.
- From line 107 to line 109, the authors wrote that the experimental procedures were approved by the Animal Care and Use Committee of Southwest University in China. Could the authors provide the number and the date of approval document related to the Animal Care and Use Committee to work with animals on this project?
- The authors should be clearer in the descriptions of materials and methods.
- The authors wrote in line 112 that mice received orally cherry extract, but in which way? Drinking water? Please, specify this at this point.
- From line 115 to line 117, the authors wrote that the doses of cherry extract were calculated by literature. Instead, why did the authors use the concentration of 300 mg/kg BW SASP?
- Line 124, Since the authors used a capital letter in the same sentence to indicate the other groups, please change (e) to (E).
- The authors could move the lines 127 to line 129, phrase “On the forth day of the feeding, the control mice showed significant symptoms of weight loss, diarrhea and bleeding in stools indicating that the UC model was developed.” In the results. The paragraph materials and methods usually doesn’t contain results.
- In the results, in the paragraph 3.2.1, in the main text the authors described figure 1c e figure 1 d before figure 1B, so the authors could change the position of each graph of figure 1, introducing each graph in the same order the authors used to mention them. This is a suggestion. Anyway, the authors have to improve the resolution of the following graphs of figure 1: 1C; 1B and 1D. For figure 1B and 1D please try to increase, if it is possible, the titles of y-axis.
- From line 204 to line 207, the caption of figure 1 should be modified. All readers should understand the main results just looking at the figure and the relative captions. The authors could write the caption indicating what the letters stand for. This could help the reader.
- Could the authors provide the scale bars in the figure 3?
- The authors should improve the quality and the resolution of figure 4 because the titles and the number on axis and the asterisks are not seen cleary.
Please pay attention to the correspondence between the main text and the figure the authors mention. See line 249, the authors mentioned that the values related to MDA are represented also in the figure 4. Instead, they are only in figure 5.
Author Response
Manuscript ID: foods-1476793
Title: Cherry polyphenol extract ameliorated dextran sodium sulfate-induced ulcerative colitis in mice by suppressing Wnt/β-catenin signaling pathway
Dear Professors,
Thank you for your useful comments and suggestions on our manuscript. We have modified the manuscript accordingly, and all changes were marked in red in the text, and the detailed response to reviewers were listed as follows:
To Reviewer 3:
This manuscript describes the protective effects of cherry polyphenol extract on Dextran sodium sulfate-induced ulcerative colitis in mice. The manuscript needs some revisions to do, as following:
- The authors should use the abbreviations only after having written them in full. See: MPO in abstact. Moreover, in line 60 the authors used the abbreviation DSS, but only in the line 67 they explained what DSS stands for. In Line 106, SPF, please write specific-pathogen-free
√ Catalase (CAT), glutathione (GSH‐Px), superoxide dismutase (SOD), malondialdehyde (MDA), myeloperoxidase (MPO) as well as nitric oxide (NO) were added in Abstract. Dextran sodium sulfate (DSS) was explained in full where it first appeared. Specific-pathogen-free was written instead of SPF in line 106. We also gave all the abbreviations in full where they first appeared in the whole manuscript.
- The authors could change the title of their manuscript, avoiding using the abbreviation DSS but writing it in full, like this “Cherry polyphenol extract ameliorated Dextran sodium sulfate -induced ulcerative colitis in mice by suppressing Wnt/β-catenin signaling pathway”. This is a suggestion that could make the title clearer also for all readers, even those not experts in this field.
√ We appreciated your suggestion, and the title was revised accordingly.
- From line 107 to line 109, the authors wrote that the experimental procedures were approved by the Animal Care and Use Committee of Southwest University in China. Could the authors provide the number and the date of approval document related to the Animal Care and Use Committee to work with animals on this project?
√The Institutional Animal Care and Use Committee (IACUC) Issue No. of Laboratory Animal Welfare and Ethics (Southwest University) is IACUC-20200315-02. We have supplemented it in Section 2.5 in the revised manuscript.
- The authors should be clearer in the descriptions of materials and methods.
√ More information was supplemented in the descriptions of materials and methods. Please check Section 2 where marked in red.
- The authors wrote in line 112 that mice received orally cherry extract, but in which way? Drinking water? Please, specify this at this point.
√The cherry polyphenol extract was first dissolved in pure water, and the concentration of the extract solution was prepared according to the body weight of mice. Then mice was fed with the extract solution through the gavage administration, and such a gavage administration was performed once a day.
More information on the way of the administration of cherry extract was supplemented in the revised manuscript.
- From line 115 to line 117, the authors wrote that the doses of cherry extract were calculated by literature. Instead, why did the authors use the concentration of 300 mg/kg BW SASP?
√ The choice of the SASP dose (300 mg/kg BW) was first based on the daily intake amount of SASP (2-3 g), Then, such a daily intake for humans was converted to the dose for mice according to the ratio (0.0026) of human body surface area to mouse body surface area, as well as the body weight of the tested mice.
- Line 124, Since the authors used a capital letter in the same sentence to indicate the other groups, please change (e) to (E).
√ It was “(L)” the low dose group instead of (E). We have deleted the “(e)”.
- The authors could move the lines 127 to line 129, phrase “On the forth day of the feeding, the control mice showed significant symptoms of weight loss, diarrhea and bleeding in stools indicating that the UC model was developed.” In the results. The paragraph materials and methods usually doesn’t contain results.
√ The phrase “On the forth day of the feeding, the control mice showed significant symptoms of weight loss, diarrhea and bleeding in stools indicating that the UC model was developed” has been moved to the Results Section 3.4.1.
- In the results, in the paragraph 3.2.1, in the main text the authors described figure 1c e figure 1 d before figure 1B, so the authors could change the position of each graph of figure 1, introducing each graph in the same order the authors used to mention them. This is a suggestion. Anyway, the authors have to improve the resolution of the following graphs of figure 1: 1C; 1B and 1D. For figure 1B and 1D please try to increase, if it is possible, the titles of y-axis.
√ The position of each graph of figure 1 was re-ordered according to the order it was described in the text. The resolution of figure 1: 1C; 1B and 1D was increased.
- From line 204 to line 207, the caption of figure 1 should be modified. All readers should understand the main results just looking at the figure and the relative captions. The authors could write the caption indicating what the letters stand for. This could help the reader.
√ The captions of all the figures were modified for purpose of understanding clearly.
- Could the authors provide the scale bars in the figure 3?
√ The scale bar is 100 μm at the magnification (×100), and the scale bar is 50 μm at the magnification (×400) in the figure 3. The scale bars were supplemented in Figure 3.
The scale bar of Figure 7 was also supplemented.
- The authors should improve the quality and the resolution of figure 4 because the titles and the number on axis and the asterisks are not seen cleary.
√ We have improved the quality and the resolution of figure 4.
- Please pay attention to the correspondence between the main text and the figure the authors mention. See line 249, the authors mentioned that the values related to MDA are represented also in the figure 4. Instead, they are only in figure 5.
√ YES, the MDA level was only presented in Figure 5, and it should not be mentioned analysis of Figure 4. We have revised it accordingly.

Round 2
Reviewer 1 Report
Authors have done a deep revision of the manuscript. Its quality and readability was considerably improved. All the suggestions were considered and all my questions were satisfactorily answered.
I have no further comments. Thus in my opinion this paper should be accepted for publication.
Author Response
Thank you for your useful comments and suggestions on our manuscript. We are very appreciated for your affirmation and advice.

Reviewer 2 Report
The authors corrected the manuscript according to the comments of the reviewers. However, I have some comments:
- The authors did not understand the comment of the reviewer. The full botanical name is an indication of the species investigated in Latin. For example, Prunus avium (L.) L.
- Table 1. Explain the principle of dividing phenolic compounds into free and bound ones.
- Table 1. Why is the tr of aglycone (quercetin) lower than tr of glycoside (quercetin 3-O-hexoside)?
- “Here, the muscular thickness is the thickness of colonic muscle. Both the thickness of colonic muscle and mucous membrane layer were closely related to the pathology of the colon. We have revised the expression in the manuscript for clearly understanding.”
I meant that the results obtained for Figure 2D (specifically, N and C, C and H) are irrelevant.
- “The total polyphenol content (TPC) of the cherry extract was about 6.84 mg GAE/g DW, here, the unit of TPC means mg of gallic acid equivalent per gram of edible part of dry cherry (mg GAE/g DW) instead of per gram of the extract.”
It does not matter the content of phenolic compounds in equivalents of gallic acid or not. According your results total content of polyphenols is less than 1% even with macroporous resin purification. Since you do not characterize the remaining 99% of extract in the manuscript, your statement is incorrect (lines 464-465).
- What is the unit of measurement – “per gram of edible part of dry cherry”? How did the authors get this “edible part”? There is no description of the method in the manuscript.
Author Response
Manuscript ID: foods-1476793
Title: Cherry polyphenol extract ameliorated dextran sodium sulfate-induced ulcerative colitis in mice by suppressing Wnt/β-catenin signaling pathway
Dear Professor,
Thank you for your useful comments and suggestions on our manuscript. We have modified the manuscript accordingly, and the detailed response to reviewers were listed as follows:
- The authors did not understand the comment of the reviewer. The full botanical name is an indication of the species investigated in Latin. For example, Prunus avium(L.) L.
√ The full botanical name of Meizao cherry used in this study is Cerasus pseudocerasus (Lindl.)G. Don, [Prunus pseudocerasus Lindl.], and we have supplemented it in the revised manuscript.
- Table 1. Explain the principle of dividing phenolic compounds into free and bound ones.
√ Phenolic compounds were commonly combined with food matrix (proteins, dietary fiber, etc.) by hydrogen bonding, hydrophobic interaction or covalent bonding, and they can be divided into the soluble free phenolics and the bound ones according to the degree of binding. The former was on the basis of the solvent-soluble extraction, and it contained both free aglycones and soluble conjugates (glycosylated forms), and the bound phenolics were mainly in the form of β-glycosides and not easy to be extracted (Sun, J.; Chu, Y.F.; Wu, X.Z.; Liu, R.H. Antioxidant and anti proliferative activities of common fruits. J. Agr. Food Chem. 2002, 50, 7449-7454. [https://doi.org/10.1021/jf0207530]). Such a principle was supplemented in the revised manuscript.
On the other hand, Section 2.2 showed the different methods for the extarction of free and bound phenolics.
- Table 1. Why is the trof aglycone (quercetin) lower than tr of glycoside (quercetin 3-O-hexoside)?
√ The tr of aglycone (quercetin) was 1.741 min, and the tr of glycoside (quercetin 3-O-hexoside) was 1.744 min. The time difference was very small, and the two compounds did not exhibite significantly retention under the current liquid phase elution conditions.
The original intention of this study was designed to acquire information on cherry polyphenols as much as possible under the same conditions through mass spectrometry technology. Therefore, a long elution time for polar components was set, and many possible phenolic compounds did appear when the elution time was at approximately 2min. However, we did not notice the difference in the tr of the two compounds.
We have consulted many published papers related to the identification of quercetin and quercetin 3-O-hexoside, and also found that the tr of aglycone (quercetin) was usually higher than the tr of glycoside (quercetin 3-O-hexoside). Therefore, it might be more reasonable that the tr of aglycone (quercetin) was 1.744 min, and the tr of glycoside (quercetin 3-O-hexoside) was 1.741 min.
Furthermore, the identification of the cherry polyphenols in this study was performed just by one method---UPLC-ESI-TOF-MS/MS, and these phenolic compounds reported in this study were identified based on the mass spectrum data and corresponding literatures. Therefore, they were only preliminarily characterized, and we have weakened the relative description in the Section 3.3 Identification of cherry polyphenol extract.
We really appreciate the serious and rigorous scientific attitude and spirit of the reviewer, and thanks for his reminding, which also gave us a choice to realize our mistake. Thank you very much!
- “Here, the muscular thickness is the thickness of colonic muscle. Both the thickness of colonic muscle and mucous membrane layer were closely related to the pathology of the colon. We have revised the expression in the manuscript for clearly understanding.”
I meant that the results obtained for Figure 2D (specifically, N and C, C and H) are irrelevant.
√ We have checked the original data of the thickness of colonic muscle layer. Admittedly, the SD values of the colonic muscle thickness were large, which might be caused by individual differences in mice.
Here, we want to emphasize that this study focused on evaluating the influence of the cherry polyphenol extract on UC mice, and such an influence was explored from not only the thickness values,but also other indexes, such as the apparent indicators (appearance, weight, colon length, colonic histopathology), and the activity of serum enzymes, the levels of inflammatory cytokines, colonic tight junction proteins, as well as the expression levels of β-catenin, c-myc, cyclin D1 and GSK-3β involved in the Wnt/β-catenin signaling pathway.
For results of Figure 2, both the intuitive picture and the thickness values of the tested colon were exhibited, the thickness values of the colonic muscle membrane layer might be regarded as an indicator or reference to help to full understand the effect of the cherry polyphenol extract on colon of UC mice to some extent, it is also the reason why we choose to show it in the manuscript. However, just as the reviewer suggested that “the results might be irrelevant”, it should not be ignored the influence of SD values on the experimental results. Therefore, we deleted the Figure 2D in the revised manuscript, and the relative description of the influence of the extract on the thickness of colonic muscle layer was also deleted.
Once again, we really appreciate the serious and rigorous scientific attitude and spirit of the reviewer, and thanks for his suggestion.
- “The total polyphenol content (TPC) of the cherry extract was about 84 mg GAE/g DW, here, the unit of TPC means mg of gallic acid equivalent per gram of edible part of dry cherry (mg GAE/g DW) instead of per gram of the extract.”
It does not matter the content of phenolic compounds in equivalents of gallic acid or not. According your results total content of polyphenols is less than 1% even with macroporous resin purification. Since you do not characterize the remaining 99% of extract in the manuscript, your statement is incorrect (lines 464-465).
√ We would like to emphasize that the “6.84 mg GAE” refered to the phenolic content of the cherry material (1g,dry weight) instead of the extract. Therefore, it might be not suitable to think that “the total content of the cherry polyphenol extract” is less than 1%”. The reason why we chose cherry material instead of the extract as the unit of the phenolic content is that it is the cherry fruit rather than its extract that people commonly eat. Therefore,it would be more instructive to people's daily diet.
On the other hand, the preparation method of the polyphenol extract used here was widely accepted. Both the methods of extraction and purification used here aim to get polyphenol compounds instead of other non- polyphenol compounds. Similarly, there are many published papers reported the bioactive capacities in mice of the polyphenol extracts prepared from fruits, vegetables, grains, and so on. For example, [1]. Flavonoid-Like Components of Peanut Stem and Leaf Extract Promote Sleep by Decreasing Neuronal Excitability, Molecular Nutrition and Food Research, https://doi.org/10.1002/mnfr.202100210; [2] Blueberry polyphenols extract as a potential prebiotic with anti-obesity effects on C57BL/6 J mice by modulating the gut microbiota, The Journal of Nutritional Biochemistry, https://doi.org/10.1016/j.jnutbio.2018.07.008; [3] A polyphenol-rich cranberry extract protects against endogenous exposure to persistent organic pollutants during weight loss in mice, Food and Chemical Toxicology, https://doi.org/10.1016/j.fct.2020.111832;[4] Polyphenol-rich vinegar extract regulates intestinal microbiota and immunity and prevents alcohol-induced inflammation in mice, Food Research International, https://doi.org/10.1016/j.foodres.2020.110064, and so on.
Of course, the reviewer`s suggestion is worth considering, that is, it is necessary to know what it is before evaluating its healthy effect. Therefore, it should be paid more attention on the characterization of the test sample.
But, from the point of giving a dietary guide, we think it is suitable to evaluate the possible functional capacities of the complex extract. Moreover, the treatment theory of traditional Chinese medicine emphasizes the compatibility of components. It is the crude/mixed extracts, not a single component or several components, that make Chinese herbal decoctions work.
- What is the unit of measurement – “per gram of edible part of dry cherry”? How did the authors get this “edible part”? There is no description of the method in the manuscript.
√ Fresh cherry were artificially enucleated to get the edible part, and the relative description was supplemented in the revised manuscript.
